# Update of a prediction model for postoperative shoulder stiffness after arthroscopic rotator cuff repair

Thomas Stojanov [1,2,3] ✉, Soheila Aghlmandi[2,4], Andreas Marc Müller[1], Philipp Moroder[5], Alexandre Lädermann[6], Cornelia Baum[1], ARCR_Pred Study Group* & Laurent Audigé [1,3,7]

## Abstract

**Background** Arthroscopic rotator cuff repair (ARCR) is a common procedure, and postoperative shoulder stiffness (POSS) is one of its most frequent adverse events, potentially necessitating individualized therapy. Our objectives were to update and internally validate a model predicting the occurrence of POSS for patients undergoing an ARCR.
**Methods** We prospectively enrolled 973 patients undergoing primary ARCR included in the ARCR_Pred dataset. A two-round Delphi survey with 53 surgeons established a consensus definition of POSS within 6 months postoperatively and a ranking of candidate prognostic factors. Treating surgeons estimated POSS risk immediately after surgery. We externally validated an existing POSS model and developed updated multivariable logistic regression models using complete-case and multiple imputed datasets.
**Results** We achieved a high consensus (88%) on the POSS definition among 44 responding shoulder surgeons, who also ranked the prognostic relevance of 71 factors for the prediction of POSS. The newly developed ARCR_Pred-POSS included 7 factors (age, acromiohumeral distance, symptom duration, baseline external rotation, active baseline abduction, baseline Oxford Shoulder Score, and surgery duration) and demonstrated superior discrimination (AUC = 0.735) and calibration (slope = 1.022) compared to the original POSS model (AUC = 0.581, slope = 0.508). Surgeons tended to overestimate the risk of POSS in their patients (AUC = 0.563, slope = 1.241).
**Conclusions** These findings support the continued development of prediction models and provide valuable outputs for optimizing surgical timing, indications, and personalized rehabilitation.

## Plain language summary

One of the most common shoulder surgeries is called arthroscopic rotator cuff repair. It helps many people recover from shoulder injuries and improves shoulder function. However, about 1 in 10 patients may experience shoulder stiffness after the surgery, which can make recovery more difficult. This study looked at ways to predict which patients are more likely to have this problem. By using data from past patients, researchers created a tool that helps doctors identify individuals at higher risk. This tool can guide decisions about when to perform surgery, who might benefit most, and how to personalize recovery plans. The results showed that these prediction tools are reliable and can help doctors make better clinical decisions, ultimately leading to improved outcomes for patients.

Clinical prediction models and their use place patients at the center of the surgical decision-making process, by providing individualized and evidence-based estimations of expected outcomes of surgical interventions[1]. In orthopedic surgery, patients expect the highest level of safety and effectiveness for their intervention. Relying on probabilities derived from prediction models, surgeons and healthcare can target specific profiles of patients at high risk of occurrence of adverse events (AE).

In this context, we initiated the ARCR_Pred study, a multicenter cohort following patients two years after an arthroscopic rotator cuff repair (ARCR), one of the main elective orthopedic procedures in Switzerland[2]. The primary objectives of the ARCR_Pred study were to develop and validate clinical prediction models for key outcomes after an ARCR, such as the expected individual shoulder function improvement and/or the occurrence of postoperative shoulder stiffness (POSS).

¹Orthopaedic Surgery and Traumatology, University Hospital of Basel, Basel, Switzerland. ²Division of Clinical Epidemiology, Department of Clinical Research, University Hospital of Basel and University of Basel, Basel, Switzerland. ³Surgical Outcome Research Center, University Hospital of Basel, Basel, Switzerland. ⁴Pediatric Research Center, University Children's Hospital Basel, Basel, Switzerland. ⁵Shoulder and Elbow Surgery, Schulthess Klinik, Zurich, Switzerland. ⁶FORE (Foundation for Research and Teaching in Orthopedics, Sports Medicine, Trauma, and Imaging in the Musculoskeletal System), Meyrin, Switzerland. ⁷Research and Development Department, Schulthess Klinik, Zurich, Switzerland. *A list of authors and their affiliations appears at the end of the paper.
✉e-mail: Thomas.stojanov@usb.ch

Depending on the definition used, POSS is one of the most frequent AE and occurs in 5 to 15% of patients within 6 months after an ARCR[3,4]. The occurrence of an early POSS is often perceived as a negative event by patients, which limits them in performing daily life activities, leading to prolonged rehabilitation or, in severe cases, to capsular release[5,6]. With the identification of patients at high risk of occurrence of POSS, recommendations for timing of surgery, surgical indications, and types of rehabilitation could be drawn based on patient individualized risk, which ultimately has the potential to maximize surgical outcomes. Yet, evidence on the specification of a prediction model for POSS after an ARCR is lacking[7,8]. A first model version was developed relying on monocentric data and showed limited predictive ability, possibly due to misspecifications[9]. We hypothesized that we could improve model performance by updating the model using solely baseline and operation variables.

The objective of the present study was to update and internally validate a model predicting the occurrence of POSS for patients undergoing an ARCR.

## Methods
### ARCR_Pred study setting
A cohort of 973 primary ARCR patients was prospectively enrolled in 18 Swiss and one German orthopedic clinics between June 2020 and November 2021 and followed up to 24 months postoperatively[2]. Ethics approval for the ARCR_Pred study was obtained on April 1st, 2020, from the lead ethics committee (EKNZ, Basel Switzerland; ID: 2019–02076, trial registration number NCT04321005). All participants provided informed written consent before enrollment in the study.

At baseline, patient demographics, psychological, socioeconomic, and clinical factors, rotator cuff integrity, concomitant local findings, operative details, and postoperative management factors were documented. Patient clinical examinations at baseline, 6, and 12 months were performed by the treating surgeon or study staff and included shoulder pain, active and passive range of motion, and strength (Constant Score). Patient-reported questionnaires at baseline, 6, 12, and 24 months were applied to determine functional scores, anxiety, and depression scores, working status, sports activities, quality of life, perception of improvement, and level of satisfaction. AEs were documented according to a consensus Core Event Set[10] and graded according to their severity[11].

### Participants follow-ups
We analyzed data from baseline forms and questionnaires, operation forms, as well as 6-month follow-up forms and questionnaires.

### Delphi process for outcome definition and prognostic factors identification
We invited the 53 shoulder surgeons who performed at least one surgery in the context of the ARCR_Pred study[2] (more details regarding the surgeon-level characteristics distribution in Supplementary Table 1) to fill out online questionnaires in the context of a two-round Delphi exercise between October 2021 and January 2022 (for more details, see Supplementary Files 1 and 2) to agree on a consensus definition of POSS and a ranking of a literature-based pre-established list of prognostic factors.

**Suggested POSS definition**. The level of agreement of participating surgeons with a consensus definition of POSS, initially developed in the context of shoulder arthroplasty, was collected[12]. POSS was defined as a composite outcome (i.e., meeting at least one of the following conditions):

1. Any restriction in passive range of motion occurring at least 3 months after the operation, implying a modification in the usual patient's care (e.g., physiotherapy, medication, manipulation under anesthesia), or:
2. A persisting restriction in passive motion in at least two planes at 6 months post-ARCR (flexion, abduction, and external rotation in zero-degree abduction). The assessment of the restriction in range of

motion has been done separately for each plane: Flexion: total motion inferior or equal to 90° or glenohumeral motion (fixed scapula) inferior or equal to 80°. Abduction: total motion inferior or equal to 80° or glenohumeral motion (fixed scapula) inferior or equal to 60°. External rotation in zero-degrees abduction: glenohumeral (fixed scapula) motion inferior or equal to 20° or inferior to 50% of the contralateral side value.

**Suggested prognostic factors for ranking**. Surgeons were also asked to rank the prognostic importance of a list of 53 potential prognostic factors on a five-item Likert scale (1: "Not important" to 5: "Highly important") relying on their clinical expertise. Of them, 22 were identified via systematic literature review[13], and 31 were identified via screening of the ARCR_Pred database by the core project team (TS, LA, CB, and AMM). Baseline, operation, and six-week variables were assessed for prognostic value, yet only baseline and operation-related prognostic factors were considered for the update and development of the present prediction model.

### Surgeon's estimation of POSS risk after surgery
After performing each surgery, surgeons were asked to estimate the probability of occurrence of POSS between 0 and 100%.

### Data management
Study data were managed using the REDCap Electronic Data Capture system[14] and exported for variable transformation (including score calculations) using Stata, version 17 (StataCorp LP, College Station, TX, USA). The statistical analysis was done using R. Additional data management and analyses steps are described in Supplementary Methods (see Supplementary Data 1).

### Statistical analysis methods
We followed the applicable TRIPOD + AI reporting recommendations (see Supplementary File 3)[15].

**Handling of missing data**. We performed and reported the whole analysis using complete case and multiple imputed data. We assumed the data were missing at random and generated ten imputed datasets using all the variables used in the various prediction models to respect the principle of congeniality[1].

**Multivariable regression modelling**. As in the first version of the model (then called KWS-POSS)[9], we used multivariable logistic regression to predict POSS occurrence with two outcome possibilities: yes ("the patient had a POSS within 6 months after the surgery") or no ("the patient did not experience a POSS within 6 months after the surgery"). Model performance was assessed using the following indicators: discrimination (AUC) and calibration (slope and intercept).

**External validation of KWS-POSS model**. In the first step, we assessed the KWS-POSS external model performance using the original set of eight baseline and operation variables with original regression coefficients: sex, overweight status, preoperative passive abduction, traumatic onset, tear severity (based on operation report), any signs of tendon degeneration, performance of an acromioplasty, and performance of a capsulotomy. We also assessed the KWS-POSS model performance after re-estimation of the coefficients on the new ARCR_Pred data.

**Model update and surgeon prediction**. In the second step, we developed and assessed the performance of new models to investigate potential improvement in model performance by using newly collected variables in the context of the ARCR_Pred study. Bootstrap validation using 500 repetitions was used to estimate overfitting. We also reported the predictive ability of performing surgeons.

**Sample size requirements.** We estimated that a seven-parameter model would minimize overfitting (less than 10%) and ensure precise estimation of key parameters in the model[16]. We fixed the number of patients at $N = 973$, an outcome prevalence of 10%, and an expected c-statistic of at least 0.73, relying on previous related findings[9].

**Model building.** Two variable selection procedures leading to seven and eight-parameter models were compared: "expert" model (the seven parameters with the highest predictive ability, as reported in the Delphi survey) and least angle selection and shrinkage operator ("Lasso") model. In our case, we used Lasso as a variable selection procedure[17]. Lasso is a linear regression method that automatically selects important variables by shrinking the coefficients of less important ones towards zero. In short, the aim of the variable selection procedure was to identify the most important prognostic factors associated with the outcome. We combined the information from both Expert-model and Lasso-model, and identified the "combined" model with seven parameters after (1) dropping variables that contained redundant information, (2) identifying the set of variables maximizing bootstrap validated performance metrics. A penalty term was then identified using the rms package, and led to the model: "combined (penalized)"[18].

**Model selection.** The apparent and bootstrap-validated performance of various models (in terms of overall performance, discrimination, intercept, and slope for calibration) was reported and compared. The model showing the best bootstrap-validated performance was selected and reported in the next steps.

**Clinical usefulness.** Clinical usefulness refers to the ability to make better decisions with the model. Usually, a cut-off is required to classify the patients as high risk (a change of treatment is indicated) or low risk (no change of routine treatment is indicated). This cut-off is then a decision threshold. In our case, the prediction model for POSS aimed to guide decisions in terms of pre-habilitation and/or advocate for more intensive and/or early rehabilitation procedures such as physiotherapy or water therapy. Four experienced surgeons (AL, AMM, CB, and PM) defined three a priori risk probability thresholds: 25%, 15%, and 10% based on the knowledge that the overall risk of POSS is around 10%.

At these three risk probability thresholds, clinical usefulness of the final prediction model was estimated by reporting the sensitivity (defined as the proportion of true positive classifications among patients with the outcome), the specificity (defined as the proportion of true negative classifications among patients without the outcome)[1].

**Model presentation.** The final model was reported in a user-friendly web-based application describing the risk of occurrence of POSS for a new patient undergoing an ARCR. The predicted probability was highlighted in green if the predicted probability was below the risk probability threshold of 10%, orange between 10 and 25%, and red above 25%.

### Reporting summary
Further information on research design is available in the Nature Portfolio Reporting Summary linked to this article.

## Results
Of the 973 patients included in the ARCR_Pred study, eight patients had data collected outside of the expected follow-up ranges. After deletion of these data, 85% of the included patients ($N = 833$) had complete data at the 6-month follow-up.

### Delphi survey results
Thirty-nine of 44 responding surgeons (88%) agreed with the suggested POSS definition. Three respondents (7%) independently suggested shifting the first outcome condition time point from 3 to 6 months. The prognostic value of 71 potential factors was assessed, 28% of which were considered for inclusion in the first KWS-POSS version of our prediction model. After reviewing the response from other surgeons, 31 surgeons (70%) decided to revise their own ratings.

### Outcome
Using the definition suggested by surgeons in the Delphi survey, 111 patients (13%) suffered from POSS, with 67 patients (60%) meeting the first condition (occurrence of symptomatic stiff shoulder leading to a modification in the routine treatment between 3- and 6-month), and 63 patients (56%) meeting the second outcome condition (persisting restrictions in range of motion at 6-month).

### Prognostic factors
In the present analysis, 41 baseline, composed of 30 patient-related and 11 diagnostic-related variables, and twelve operation variables were identified in the Delphi process and individually described. Distribution, percentage of missing values, prognostic value for the prediction of the outcome (using the Delphi results), and univariable associations with the outcome were reported (see Supplementary Data 2 and Supplementary Tables 2 and 3).

### External model validation
Using the original regression coefficients, the model KWS-POSS showed poor discrimination (AUC = 0.581) and calibration indicators (Intercept = −1.269 and Slope = 0.508). After re-estimation of the coefficients on the ARCR_Pred data, the model improved in terms of apparent discrimination (AUC = 0.637), but performance dropped after bootstrap validation (AUC = 0.587, Intercept = −0.564, Slope = 0.686). Surgeons overestimated the risks of POSS for their own patients (Slope = 1.241) and showed no discriminative ability (AUC = 0.563) (Table 1).

### Model development, specification, and performance
Performance metrics are available for each model (Table 1). The "expert" model was defined using the seven-parameters with the highest mean prognostic value rating: baseline shoulder stiffness, diabetes status, baseline Oxford Shoulder Score, baseline level of depression and anxiety, performance of a capsulotomy, symptom duration, and tear etiology. The Lasso procedure excluded tear etiology from that model. The "Lasso" procedure selected the 8 following parameters: acromiohumeral distance, age at surgery, baseline Oxford Shoulder Score, number of anchors used during the surgery, surgery duration, baseline active abduction (from the affected side), baseline passive abduction (glenohumeral—fixed scapula from the contralateral side), baseline external rotation at 0° abduction (glenohumeral passive - fixed scapula - affected side). The "combined" model was defined using the following seven parameters: baseline external rotation (affected side), baseline active abduction (affected side), baseline Oxford Shoulder Score, symptom duration, age at surgery, acromiohumeral distance, and operation duration. After penalization, the combined ARCR_Pred model was similar in terms of model performance on complete-case and multiple imputed data and had limited overfitting (Table 2).

### Clinical usefulness
The combined ARCR_Pred-POSS model showed positive net benefit over the "treat all" and "treat none" strategies for the whole range of predicted probabilities (Fig. 1), even after bootstrap validation, while the surgeon prediction was similar to a "treat all" strategy. The sensitivity of the prediction model increased (31, 57, and 85%) with a decrease in the risk probability threshold (25, 15, and 10%), and the specificity decreased (91, 73, and 52%, respectively) (Table 3).

### Model presentation
The model is presented in Table 4 and was embedded into a web-based responsive app, highlighting the predicted probability of occurrence of POSS (https://arcrpred.shinyapps.io/ARCR_Pred-POSS-Logistic-Model/).

**Table 1 | Model performance comparison**

| Model | Apparent | | | Bootstrap validation[c] | | |
|---|---|---|---|---|---|---|
| | Discrimination | Intercept | Slope | Discrimination | Intercept | Slope |
| Model KWS-POSS (original) | 0.582 | −1.269 | 0.508 | . | . | . |
| Model KWS-POSS (recalibrated) | 0.637 | 0 | 1 | 0.587 | −0.565 | 0.685 |
| Model KWS-POSS (recalibrated) \| Multiple imputation[a] | 0.623 | 0 | 1 | 0.576 | −0.592 | 0.674 |
| Surgeon prediction after surgery | 0.563 | −2.239 | 1.241 | . | . | . |
| Model ARCR_Pred – Expert | 0.669 | 0 | 1 | 0.648 | −0.212 | 0.878 |
| Model ARCR_Pred - Expert \| Multiple imputation[a] | 0.661 | 0 | 1 | 0.64 | −0.201 | 0.882 |
| Model ARCR_Pred – LASSO | 0.733 | 0 | 1 | 0.711 | −0.178 | 0.892 |
| Model ARCR_Pred - LASSO \| Multiple imputation[a] | 0.661 | 0 | 1 | 0.64 | −0.201 | 0.882 |
| Model ARCR_Pred - Expert (reduced via LASSO) | 0.67 | 0 | 1 | 0.654 | −0.162 | 0.906 |
| Model ARCR_Pred - Expert (reduced via LASSO) \| Multiple imputation[a] | 0.662 | 0 | 1 | 0.647 | −0.155 | 0.91 |
| Model ARCR_Pred – Combined | 0.735 | 0 | 1 | 0.716 | −0.161 | 0.906 |
| Model ARCR_Pred - Combined \| Multiple imputation[a] | 0.722 | 0 | 1 | 0.707 | −0.125 | 0.927 |
| Model ARCR_Pred - Combined (penalized) | 0.735 | 0 | 1 | 0.718 | 0.035 | 1.022 |
| Model ARCR_Pred - Combined (penalized)[b] \| Multiple imputation[a] | 0.724 | 0 | 1 | 0.709 | 0.032 | 1.022 |

[a]Multiple imputation was performed using 10 datasets, reported model performance is the average performance across the 10 multiple imputed datasets.
[b]The optimal penalty term was identified on each multiple imputed dataset and was averaged.
[c]Bootstrap validation was performed only for newly developed prediction models using $B = 500$ repetitions for complete-case data, and $B = 50$ repetitions for multiple imputed data (hence leading to 50 repetitions * 10 datasets = 500 repetitions).

**Table 2 | Summary of ARCR_Pred-POSS model performance metrics**

| Aspect | Measure | Complete case | | Multiple imputation | |
|---|---|---|---|---|---|
| | | Apparent ($N = 833$) | Bootstrap validation ($B = 500$) | Apparent ($M = 10$) | Bootstrap validation ($B = 50$) |
| Overall performance | $R^2$ | 13% | 12% | 12% | 11% |
| Discrimination | AUC | 0.735 | 0.718 | 0.718 | 0.702 |
| Calibration | Intercept | .. | 0.035 | .. | 0.012 |
| | Slope | .. | 1.02 | .. | 1.02 |

AUC area under the receiver operating characteristics curve.
This table shows the apparent and bootstrap-validated model performance metrics of the ARCR_Pred-POSS combined model after application of a penalty term. A sensitivity analysis was conducted to compare performance metrics on multiple imputed data. N represents the number of patients in the complete case analysis. B represents the number of bootstraps. M represents the number of imputed datasets.

## Discussion

Based on a consensus definition of postoperative shoulder stiffness (POSS) and a ranking of key prognostic factors, we developed a prediction model, ARCR_Pred-POSS, which outperformed surgeons and an already existing model in predicting the risk of POSS within 6 months after an ARCR. Our model, incorporating seven baseline and operative variables, showed good discrimination and calibration, and was made accessible through a user-friendly app for educational and presentation purposes.

The rationale for the development of new prediction models was the lack of discriminative ability of the KWS-POSS[9] model on the ARCR_Pred data (AUC = 0.58). Surgeons overestimated the risk of POSS for their own patients (slope = 1.24). Surgeons had almost no discriminative ability in predicting POSS (AUC = 0.56). The final model ARCR_Pred-POSS was composed of seven baseline and operation variables, showing relatively good performance on both complete-case (AUC = 0.735) and multiple imputed datasets (AUC = 0.724), also after bootstrap validation (AUC = 0.718 and AUC = 0.709, respectively). These metrics indicate small overfitting and good potential for prediction accuracy in new patients. A penalty term improved the calibration of the model.

The identification of factors specifying the prediction model has important clinical implications. For instance, a reduced acromio-humeral distance may indicate larger degenerative tears[19–21], suggesting

the need for more complex reconstruction methods beyond ARCR. Non-modifiable factors, such as age, should be considered when aiming to prevent the occurrence of postoperative shoulder stiffness (POSS). This finding is in line with the results of two studies included in our recent review, synthesizing the evidence related to prognostic factors for POSS[4]. Repairs of smaller, partial, and traumatic tears—which occur more frequently in younger patients—have been shown to be associated with higher rates of POSS in previous studies[22]. High incidence of pre-operative rotator interval fibrosis observed in partial tears might partly explain this puzzling association[23]. Additionally, younger patients are more often affected by traumatic lesions, which are associated with early intramuscular edema formation. Recent experimental studies have demonstrated that this trauma-related edema leads to muscle fiber disruption and progressive interstitial fibrosis, which may also contribute to the higher rates of postoperative shoulder stiffness observed in this population[24]. Partially modifiable factors, including surgery duration, symptom duration, and baseline functional scores, can potentially be optimized to improve outcomes. While surgery should ideally be scheduled early when clearly indicated, the optimal timing for surgery remains uncertain and must balance the benefits of prehabilitation to maximize preoperative functional scores. Interestingly, the number of anchors used and surgery duration emerged as a factor associated with

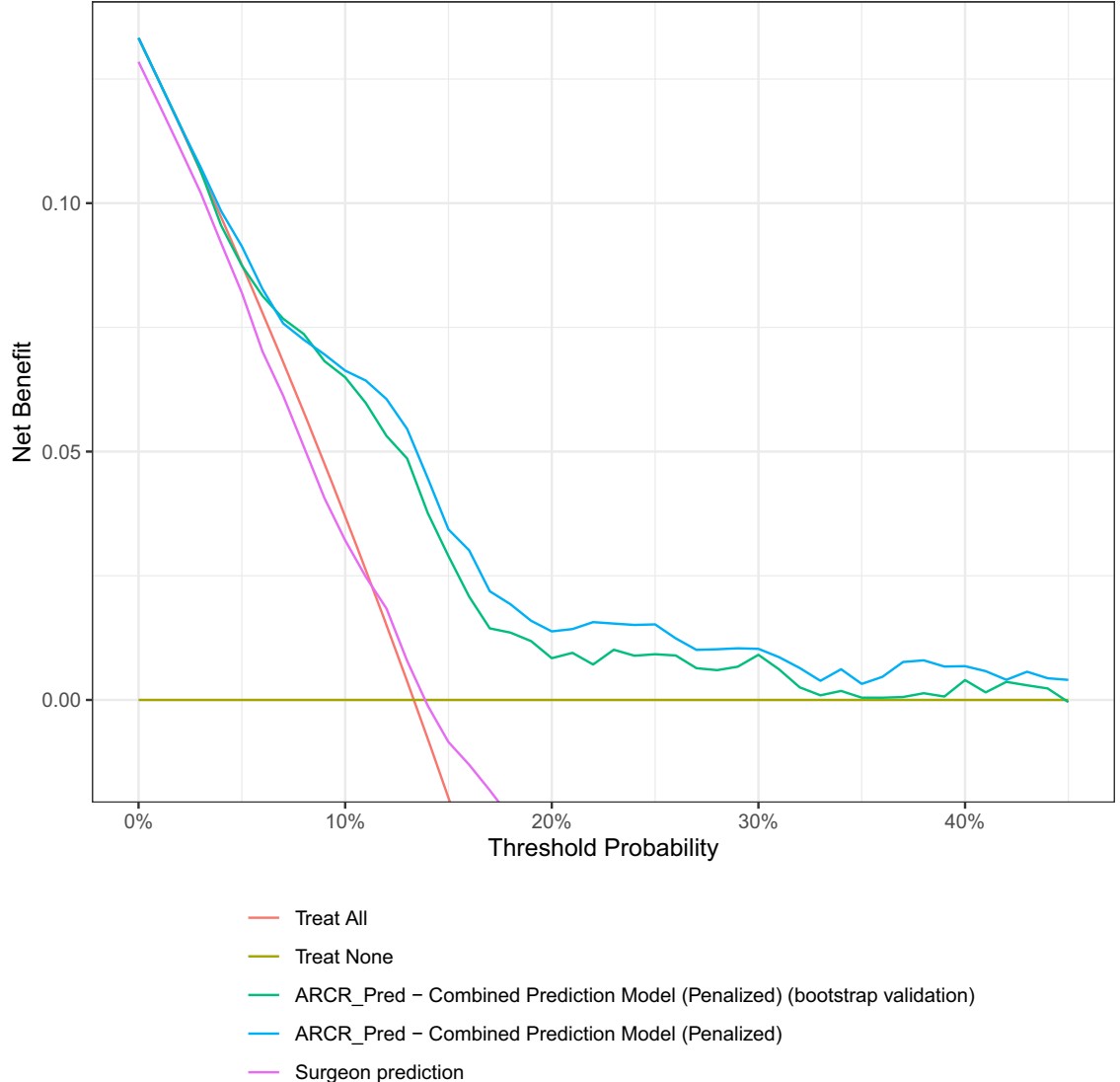

**Fig. 1 | Clinical usefulness of the ARCR_Pred-POSS model. Note**. This figure shows the clinical usefulness of the ARCR_Pred-POSS model before (in green) and after bootstrap validation (in blue) in comparison to the surgeon prediction (in pink) and to the "treat all" (in red) and "treat none" (in dark green) strategies, over a range of risk probability thresholds (0 to 40%). The ARCR_Pred-POSS model showed superiority for clinical usefulness at various risk probability thresholds.

worse outcome using our Lasso selection procedure. This finding is also supported by another study for repair integrity outcome[25]. Surgery duration is likely a proxy variable reflecting its association with other factors that influence the occurrence of POSS.

Our findings support a transparent and evidence-based treatment decision-making process, where individualized expected outcomes can be shared with patients. We defined three risk probability groups (0–10%: low risk, 10–25%: moderate risk, and 25–100%: high risk). For example, moderate and high-risk patients could be suggested alternative and preventive rehabilitation protocols, such as balneotherapy, recently shown to facilitate the early post-operative range of motion recovery[26].

These risk probability thresholds were not unanimously agreed upon by all clinicians involved in the project. They should therefore be considered illustrative at this stage. Their relevance and applicability will be further assessed and validated once the prediction models are implemented in clinical practice.

A user-friendly, responsive web-based app was created for educational and demonstrative purposes and is publicly available.

While the identification of prediction models improving surgical decision-making is a topic of rising interest in orthopedics[27], these should be implemented in clinical practice to drive effective changes for

**Table 3 | Sensitivity and specificity of the ARCR_Pred-POSS prediction model at various risk probability thresholds**

| Risk probability threshold | ARCR_Pred-POSS model | POSS | No POSS | Total | Sensitivity/ Specificity |
|---|---|---|---|---|---|
| 25% | *Predicted +* | 34 | 64 | 98 | 31% / 91% |
| | *Predicted −* | 77 | 658 | 735 | |
| 15% | *Predicted +* | 63 | 195 | 258 | 57% / 73% |
| | *Predicted −* | 48 | 527 | 575 | |
| 10% | *Predicted +* | 94 | 349 | 443 | 85% / 52% |
| | *Predicted −* | 17 | 373 | 390 | |
| | Total | 111 | 722 | 833 | |

This table shows the predictive performance of the ARCR_Pred-POSS model across different risk probability thresholds. For each threshold (25%, 15%, and 10%), the table shows the number of events and non-events correctly predicted by the model. As the threshold decreases, the number of predicted events increases while the number of predicted non-events decreases, reflecting the trade-off between sensitivity and specificity in the model's predictions.
*POSS* post-operative shoulder stiffness, *Predicted+* The model predicted an event for the given patient, Predicted− The model predicted a non-event for the given patient.

## Table 4 | Multivariable ARCR_Pred-POSS model

| ARCR_Pred-POSS model (*N* = 7) | OR (95% CI) | *p* value |
|---|---|---|
| **Baseline variables (*N* = 6)** | | |
| **Sociodemographic factors (*N* = 1)** | | |
| Age at surgery (in years) | 0.98 (0.95–1.00) | 0.059 |
| **Diagnostic-related factors (*N* = 2)** | | |
| Acromiohumeral distance (in mm) | 0.87 (0.77–0.93) | 0.002 |
| Symptom duration of more than 6 months | 1.69 (1.26–3.14) | 0.025 |
| **Pre-operative scores (*N* = 3)** | | |
| Abduction (active - affected side) (10-degree unit) | 0.93 (0.87–0.98) | 0.018 |
| External rotation at 0° abduction (active - affected side) (10° unit) | 0.91 (0.80–1.03) | 0.13 |
| Oxford Shoulder Score (0–48) | 0.96 (0.93–0.98) | 0.002 |
| **Operation variable (*N* = 1)** | | |
| Operation duration (10-min unit) | 1.05 (0.99–1.11) | 0.12 |

*CI* confidence interval, *OR* odds ratio.
This table presents results from the ARCR_Pred-POSS model, based on seven baseline and operation variables. The model examines the associations between the variables retained in the final model and the occurrence of post-operative shoulder stiffness.

patients. Such an app would profit after certification from a direct integration within an electronic health record system, hence supporting the decision-making on a day-to-day basis. Moreover, recent research highlighted the existence of models that could be continuously updated using Bayesian dynamic modelling[28]. The efficacy of clinical prediction model tools should be assessed using a randomized controlled trial. Recent research in the field of arthroplasty showed that the use of artificial intelligence-enabled decision aid tools ultimately improved patient satisfaction with regards to decision quality and level of shared decision making[29,30]. These findings were also supported by the results of a recent review reporting that healthcare professionals and patients perceptions in North America were mostly positive towards the utilization of artificial intelligence-enabled risk prediction models[31]. Yet, review authors reminded that transparency and healthcare and patients preferences should be considered, a statement supported by European researchers in the field of outcome prediction in surgery[32,33]. Further research on the implementation and efficacy of clinical prediction models' tools is needed.

The ARCR_Pred-POSS model was developed and validated using multicenter data from 19 Swiss and German orthopedic clinics, with a case-mix in baseline scores severity, and in patient profiles[2]. While we assessed the selection bias of the ARCR_Pred study elsewhere[34], ARCR_Pred study patients were recruited and operated during the COVID-19 pandemic. This specific patient selection might limit the external validity of the ARCR_Pred-POSS model, which would remain to be assessed in an external cohort. Additionally, participants from the same study center are likely to have been subject to similar healthcare processes, and in our case, surgical techniques, leading to correlations between observations within centers and "clustering". The ARCR_Pred-POSS model performance at a cluster level remains to be assessed. Indeed, prediction model performance usually varies across clusters. Research guidance already exists in this area[35]. In the field of ARCR, analysis of performance of prediction models at a study center level was, to the best of our knowledge, not yet performed.

In the present study, we used a logistic regression model on a limited sample size. Recent studies using artificial intelligence models for the prediction of occurrence of AEs following an ARCR showed better model performance[36]. However, in the medical literature, the relative contribution of artificial intelligence models in comparison to regression models for outcome prediction tasks remains controversial[37]. More research identifying optimal prediction models for outcome prediction in surgical patients is needed. Of note, the variables suggested in the Delphi survey for the assessment of prognostic relevance were identified using the findings of our systematic review[4]. While seeking the surgeons' expertise for the initial variable selection was essential, the perspectives of patients and rehabilitation professionals were not considered at that stage. Doing so may have resulted in a different variable selection for the prediction model. In addition, the final model relied on shoulder surgeon expert knowledge and Lasso variable selection procedure only. The Lasso, while effective for variable selection, may exclude variables with small but potentially clinically relevant associations[17], hence leaving room for further methodological comparison across modern variable selection procedures, such as Bayesian predictive projection[38].

## Conclusions

This study established a consensual approach among orthopedic surgeons for the development of a prediction model for POSS within 6 months following an ARCR. The developed model, consisting of six baseline parameters and surgery duration, demonstrates superior performance in comparison to existing models. Furthermore, our data demonstrated that the treating orthopedic surgeons are unable to predict the risk of POSS. With consensually defined risk probability groups and a user-friendly web app interface, clinicians, healthcare, and patients are thought to make informed decisions based on evidence. Yet, benefits of the use of such prediction models on concrete patient outcomes remain an important area of research for orthopedic patients.

## Data availability

Following a period of embargo of at least two years after the end of the study in September 2024, metadata describing the type, size, and content of the dataset will be published along with the study protocol on the open repository Zenodo (https://zenodo.org/). Researchers wishing to access the full dataset will be able to file a request with the Data Access Committee of the Medical Faculty of the University of Basel (MF-DAC – email: med-dac@unibas.ch). The MF-DAC will act as an independent assessor of the request and grant access to the dataset if all ethical, legal, and scientific conditions are met.

## Code availability

The statistical code produced to obtain the results was uploaded to Zenodo[39], and will be made available upon request.

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

## Acknowledgements

This project is funded by the Swiss National Science Foundation (SNF Project ID 320030_184959, http://p3.snf.ch/project-184959).

## Authors contributions

Thomas Stojanov (T.S.): Data curation (equal); Formal analysis (equal); Investigation (equal); Methodology (equal); Software (lead); Validation (supporting); Visualization (lead); Writing—original draft (lead); Writing—review and editing (lead). Laurent Audigé (L.A.): Conceptualization (equal); Data curation (equal); Formal analysis (equal); Funding acquisition (equal); Investigation (equal); Methodology (equal); Project administration (equal); Resources (equal); Supervision (equal); Writing—original draft (supporting). Andreas Marc Müller (A.M.M.): Conceptualization (equal); Funding acquisition (equal); Project administration (equal); Resources (equal); Supervision (equal); Writing—review and editing (equal). Soheila Aghlmandi (S.A.): Formal analysis (equal); Methodology (equal); Supervision (supporting); Validation (lead); Writing—original draft (supporting). Philipp Moroder (P.M.): Writing—review and editing (equal). Alexandre Lädermann (A.L.): Writing— review and editing (equal). Cornelia Baum (C.B.): Writing—review and editing (equal). ARCR_Pred Study Group: Investigation (supporting).

## Competing interests

The authors declare no competing interests.

## Additional information

## ARCR_Pred Study Group

Claudio Rosso[8], Philipp Moroder[9], Doruk Akgün[9], Isabella Weiss[9], Eduardo Samaniego[9], Thomas Suter[10], Sebastian A. Müller[10], Markus Saner[10], Claudia Haag-Schumacher[10], Mai Lan Dao Trong[11], Carlos Buitrago-Tellez[11], Julian Hasler[11], Ulf Riede[11], Beat Moor[12], Matthias Biner[12], Nicolas Gallusser[12], Christoph Spormann[13], Britta Hansen[13], Holger Durchholz[14], Gregory Cunningham[15], Alexandre Lädermann[6,16], Michael Schär[17], Rainer Egli[17], Stephanie Erdbrink[17], Kate Gerber[17], Paolo Lombardo[17], Johannes Weihs[17], Matthias Flury[18], Ralph Berther[18], Christine Ehrmann[18], Larissa Hübscher[18], David Schwappach[19], Karim Eid[20], Susanne Bensler[20], Yannick Fritz[20], Emanuel Benninger[21], Philemon Grimm[21], Markus Pisan[21], Markus Scheibel[22], Laurent Audigé[22], Daniela Brune[22], Marije de Jong[22], Stefan Diermayr[22], Marco Etter[22], Florian Freislederer[22], Michael Glanzmann[22], Cécile Grobet[22], Christian Jung[22], Fabrizio Moro[22], Ralph Ringer[22], Jan Schätz[22], Hans-Kaspar Schwyzer[22], Martina Wehrli[22], Barbara Wirth[22], Christian Candrian[23], Filippo Del Grande[23], Pietro Feltri[23], Giuseppe Filardo[23], Francesco Marbach[23], Florian Schönweger[23], Bernhard Jost[24], Michael Badulescu[24], Stephanie Lüscher[24], Fabian Napieralski[24], Lena Öhrström[24], Martin Olach[24], Jan Rechsteiner[24], Jörg Scheler[24], Christian Spross[24], Vilijam Zdravkovic[24], Matthias A. Zumstein[25], Annabel Hayoz[25], Julia Müller-Lebschi[25], Karl Wieser[26], Paul Borbas[26], Samy Bouaicha[26], Roland Camenzind[26], Sabrina Catanzaro[26], Christian Gerber[26], Florian Grubhofer[26], Anita Hasler[26], Bettina Hochreiter[26], Roy Marcus[26], Farah Selman[26], Reto Sutter[26], Sabine Wyss[26], Christian Appenzeller-Herzog[27], Andreas Marc Müller[28], Soheila Aghlmandi[28], Cornelia Baum[28], Franziska Eckers[28], Kushtrim Grezda[28], Simone Hatz[28], Sabina Hunziker[28], Thomas Stojanov[28], Mohy Taha[28] & Giorgio Tamborrini-Schütz[28]

[8]ARTHRO Medics, Basel, CH (ART), Switzerland. [9]Charitè Medicine University, Berlin, DE (BER), Germany. [10]Cantonal Hospital Baselland, Bruderholz, CH (BRU), Switzerland. [11]Public Hospital Solothurn, Solothurn, CH (BSS), Switzerland. [12]Hôpital du Valais –Centre Hospitalier du Valais Romand, Martigny, CH (CHV), Switzerland. [13]Endoclinic, Zurich, CH (END), Switzerland. [14]Klinik Gut, St Moritz, CH (GUT), Switzerland. [15]Hirslanden Clinique la Colline, Geneva, CH (HIR), Switzerland. [16]La Tour Hospital, Meyrin, CH (HUG, FORE), Switzerland. [17]Inselspital, Bern, CH (INB), Switzerland. [18]In-Motion, Wallisellen, CH (INM), Switzerland. [19]Institute of Social and Preventive Medicine (ISPM), University Bern, Bern, CH, Switzerland. [20]Cantonal Hospital Baden, Baden, CH (KSB), Germany. [21]Cantonal Hospital Winterthur, Winterthur, CH (KSW), Switzerland. [22]Schulthess Klinik, Zurich, CH (KWS), Switzerland. [23]Ospedale Regionale di Lugano, Lugano, CH (LUG), Switzerland. [24]Cantonal Hospital St. Gallen, St. Gallen, CH (SGA), Switzerland. [25]Orthopädie Sonnenhof, Bern, CH (SON), Switzerland. [26]University Clinic Balgrist, Zurich, CH (UKB), Switzerland. [27]University Library Basel, University Basel, Basel, CH, Switzerland. [28]University Hospital Basel, Basel, CH (USB), Switzerland.

