## [Transparent Peer Review file · Communications Medicine]

Update of a prediction model for postoperative shoulder stiffness after arthroscopic rotator cuff repair

Corresponding Author: Mr Thomas Stojanov

Version 0:

Reviewer comments:

Reviewer #1

(Remarks to the Author)

Based on multicenter cohort data and Delphi method consensus, this study updates and validates the prediction model of shoulder stiffness after ARCR (POSS), with great clear clinical significance and methodological innovation. The ARCR Pred-POSS model was developed by combining expert opinion and statistical methods (such as Lasso regression). The final ARCR_PreD-POSS model was superior to the existing models and the subjective prediction of surgeons in discrimination (AUC=0.735) and calibration (slope =1.022). The clinical utility is enhanced through a user-friendly web application presentation model. The study design is standard, the data transparency is high, and the statistical analysis is comprehensive. However, the following issues should be noted before publication:

Abstract :

Appropriate.

Background:

Brief and appropriate.

Method:

Line 122: recommend adding description about the details of surgeon (surgical number per year, experience, etc.), which may also be added into multivariable regression analysis.

Line 162-165: Do you perform the sensitivity analysis and how much data were missing? You should perform sensitivity analysis to make sure the imputation is correct.

Line 216 were the thresholds only defined by experienced surgeons? Is there any study discussing about the classification?

Results :

Line 242-245, why 63 patients account for 40%?

Table3- The final model included an OR of 1.05 (p=0.12) for the operation duration, but its clinical significance was not explained (e.g., whether the duration of the procedure reflected surgical complexity or operator experience)

Discussion:

Appropriate.

Reviewer #2

(Remarks to the Author)

This is an interesting paper on a clinically important subject. Predictive models are difficult to construct, often ending up with R²-values from 10-15, with the model thus only explaining some of the variance. The authors have however made a good job in trying to improve their original model, using sound methods. The resulting model seem usable in a clinical context.

Abstract

Lines 66-67: The abbreviation "the original KWS-POSS model" is not explained in the abstract. Authors claim superior discrimination and calibration, but the reader does not get measures for the original POSS model. Does the word limit for abstracts in the journal preclude an explanation?

Introduction

Lines 87-88: Postoperative stiffness in the first 3-6 months after ARCR is common and usually the stiffness subsides, and most patients seem to end up with satisfactory range of motion. There also seem to be an association with post-operative stiffness and healing, with lower rates of retear in patients with postoperative stiffness (see for example Takahashi et al. *Arthroscopy*, Vol 40, No 8 (August), 2024: pp 2186-2194). The authors could elaborate a bit on why it is important to assess risk of stiffness in the preoperative setting. Why is this study really needed? And what is the natural course of postoperative stiffness after ARCR? As I understand the reasoning, and according to clinical experience, there is often some stiffness at 3 months (especially in those that have healed) but at 6 months most patients have regained motion. It would increase interpretability if some of these aspects were clarified for the reader.

Methods

Lines 104-105: patient inclusion and operations were during the Covid pandemic. It is possible that this may have influenced patient selection for operative treatment, thereby possibly influencing the external validity of the results. Could patients more prone for stiffness be less represented in this cohort? This could be brought up in the discussion.

Lines 107-111. Who did the scoring, a person not otherwise involved in treatment or clinical follow-up?

Line 172: AUC and Calibration are good and recommended indicators. Other studies on predictive models also use the Brier score as well. This way you get a better impression of the overall performance of the predictive model (see fx Steyerberg EW, Vickers AJ, Cook NR, Gerds T, Gonen M, Obuchowski N, et al. Assessing the performance of prediction models: a framework for traditional and novel measures. *Epidemiology*. 2010 Jan;21(1):128–38.). Adding the Brier score could thus ease comparison with other models. This is just a suggestion and not mandatory to accept the manuscript for publication.

Line 197: A brief description of the Lasso method would enhance easy understanding.

Results

Line 265: I think eTable 5 is highly interesting in the way the reader can follow the model development and the improvements in performance. Why not put it in the article, not only in Supplementary material?

Discussion

Good discussion, especially bringing up Bayesian methods and machine-learning methods. I am curious about what the authors think are the main reasons for the improvement in model performance, from the original KWS-POSS model to this final model. What is the contribution of the larger sample size and the expert consensus?

Reviewer #3

(Remarks to the Author)

Comments to the Author

The article is about updating and validating a prediction model for postoperative shoulder stiffness (POSS) after arthroscopic rotator cuff repair. The study is based on 973 patients from multicenter, following them up for as long as 2 years. The results show that the model includes six baseline parameters and operation duration, and it performs better than existing models. This article has made a valuable contribution to predicting the risk of POSS. However, there are several shortcomings in the article regarding the transparency of parameter selection, explanation of parameters, rationality of follow-up duration, and clinical applicability of the model. By further improving these aspects, the scientific and practical nature of the article can be enhanced, making it more likely to be widely accepted and applied in clinical practice.

Review Comment 1: Transparency of Parameter Selection

Question:

Although the article mentions using methods such as Delphi surveys and Lasso regression to select parameters, there is a lack of detailed explanation for the final inclusion of certain parameters. For example, in Table 3, which presents the core results, the statistical significance of the "age" and "operation duration" parameter are above 0.05, yet it is still included in the model. This may make it difficult for readers to understand the reason for its inclusion.

Suggestion:

The authors should provide a more detailed account of the parameter selection process and the rationale for the final inclusion in the model. Additionally, the limitations of the parameter selection methods could be discussed, along with possible directions for improvement.

Review Comment 2: Explanation of the Age Parameter

Question:

The model includes the "age" parameter, which has an OR value of 0.98 ($p = 0.059$), indicating that the risk of POSS slightly decreases with each additional year of age. This result contradicts clinical intuition, as it is generally believed that older age is associated with a higher risk of POSS. The authors need to provide a detailed explanation for this finding.

Suggestion:

Discuss the potential physiological and pathological mechanisms underlying the relationship between age and POSS risk. For instance, could it be that older patients have weaker inflammatory responses or better rehabilitation compliance, thereby reducing the risk of POSS?

Review Comment 3: Rationality of Follow-up Duration

Question:

The study followed up for 2 years, but the main objective was to predict the risk of POSS within 6 months after surgery.

Although a 2-year follow-up can provide more comprehensive data, it may increase the complexity and cost of the study, and its contribution to the primary goal (predicting POSS risk within 6 months) is not clear.

Suggestion:

The authors should provide a detailed discussion on the necessity of a 2-year follow-up, including plans for utilizing the long-term data and its potential clinical significance. If the 2-year follow-up is mainly to capture long-term complications, this should be clearly stated in the discussion section, along with suggestions on how to use this data to further optimize clinical decision-making.

Review Comment 4: Clinical Applicability of the Model

Question:

Although the model performs well in terms of discrimination and calibration, the article lacks a detailed discussion on the clinical applicability of the model. For example, how can the model be applied in clinical practice, and how can clinical decisions be adjusted based on the model's results?

Suggestion:

Risk Probability Group Thresholds: In the discussion section, the authors should provide a detailed explanation of the rationale behind the thresholds for the three risk probability groups (0 - 10%: low risk, 10 - 25%: moderate risk, 25% - 100%: high risk), including whether they are based on clinical experience or data-driven analysis.

Enhanced Clinical Applicability Discussion: Include a more detailed discussion on the clinical applicability of the model, such as its application in real clinical scenarios, potential clinical impact, and how it can be integrated with existing clinical guidelines. For example, for high-risk patients, are more aggressive preoperative rehabilitation plans or postoperative interventions recommended?

Version 1:

Reviewer comments:

Reviewer #2

(Remarks to the Author)

Dear Authors and Editors,

I have read the revised manuscript and it is improved enough for publication in my opinion. Congratulations on a well-performed study.

Br

Mats Ranebo, MD PhD

Reviewer #3

(Remarks to the Author)

This manuscript titled "Update of a prediction model for postoperative shoulder stiffness after arthroscopic rotator cuff repair" presents a well-designed and methodologically sound study that updates and validates a prediction model for postoperative shoulder stiffness (POSS) after arthroscopic rotator cuff repair (ARCR). The authors used a multicenter prospective cohort design with a large sample size and followed TRIPOD+AI reporting recommendations. The new ARCR_Pred-POSS model shows superior performance in discrimination and calibration compared to the original model and surgeon predictions. The study is well-conducted, and the results have significant clinical implications. I recommend accepting this manuscript for publication.

Point-by-point responses – Communications Medicine

Manuscript no.: **COMMSMED-24-0678A**

Manuscript title: **Update of a prediction model for postoperative shoulder stiffness after arthroscopic rotator cuff repair**

Authors: Thomas Stojanov, PhD^{1,2*}, Soheila Aghlmandi, PhD^{2,3}, Andreas Marc Müller, MD¹, Philipp Moroder, MD⁴, Alexandre Lädermann, MD⁵, Cornelia Baum, MD¹, ARCR_Pred Study Group*, Laurent Audigé, PhD^{1,4,6}

Contents

Overall comments	2
Abstract	3
Introduction	4
Methods	5
Results	8
Discussion	8
References.....	11

Overall comments

Reviewer #1: Based on multicenter cohort data and Delphi method consensus, this study updates and validates the prediction model of shoulder stiffness after ARCR (POSS), with great clear clinical significance and methodological innovation. The ARCR Pred-POSS model was developed by combining expert opinion and statistical methods (such as Lasso regression). The final ARCR_PreD-POSS model was superior to the existing models and the subjective prediction of surgeons in discrimination (AUC=0.735) and calibration (slope =1.022). The clinical utility is enhanced through a user-friendly web application presentation model. The study design is standard, the data transparency is high, and the statistical analysis is comprehensive.

Reviewer #2: This is an interesting paper on a clinically important subject. Predictive models are difficult to construct, often ending up with R²-values from 10-15, with the model thus only explaining some of the variance. The authors have however made a good job in trying to improve their original model, using sound methods. The resulting model seem usable in a clinical context.

Reviewer #3: The article is about updating and validating a prediction model for postoperative shoulder stiffness (POSS) after arthroscopic rotator cuff repair. The study is based on 973 patients from multicenter, following them up for as long as 2 years. The results show that the model includes six baseline parameters and operation duration, and it performs better than existing models. This article has made a valuable contribution to predicting the risk of POSS. However, there are several shortcomings in the article regarding the transparency of parameter selection, explanation of parameters, rationality of follow-up duration, and clinical applicability of the model. By further improving these aspects, the scientific and practical nature of the article can be enhanced, making it more likely to be widely accepted and applied in clinical practice.

Authors response:

We would like to greatly thank the reviewer for their detailed review and comments, and obviously for their positive feedback towards our work. We are happy to provide the reviewers with detailed point-by-point responses for each of their comment in the following pages. We re-organized the comments section by section and we also provide here the changes we brought to the manuscript. The line number we are referring to in the present document are the line numbers of the new clean manuscript version.

Abstract

Reviewer #2: Lines 66-67: The abbreviation “the original KWS-POSS model” is not explained in the abstract. Authors claim superior discrimination and calibration, but the reader does not get measures for the original POSS model. Does the word limit for abstracts in the journal preclude an explanation?

Authors response: We agree with the reviewer’s comment, and we provided more details towards the KWS-POSS model and the surgeon predictions.

Suggested changes: “Based on these results, and using the data of all the patients included in the ARCR_Pred study (N = 973 ARCR), the newly developed ARCR_Pred-POSS model demonstrated superior discrimination (AUC = 0.735) and calibration (slope = 1.022) compared to the initial version of the POSS model (AUC = 0.581, slope = 0.508). Treating surgeons overestimated the risk of POSS in their patients (AUC = 0.563, slope = 1.241).”

Introduction

Reviewer #2: Lines 87-88: Postoperative stiffness in the first 3-6 months after ARCR is common and usually the stiffness subsides, and most patients seem to end up with satisfactory range of motion. There also seem to be an association with post-operative stiffness and healing, with lower rates of re-ear in patients with postoperative stiffness (see for example Takahashi et al. Arthroscopy, Vol 40, No 8 (August), 2024: pp 2186-2194). The authors could elaborate a bit on why it is important to assess risk of stiffness in the preoperative setting. Why is this study really needed? And what is the natural course of postoperative stiffness after ARCR? As I understand the reasoning, and according to clinical experience, there is often some stiffness at 3 months (especially in those that have healed) but at 6 months most patients have regained motion. It would increase interpretability if some of these aspects were clarified for the reader.

Authors response: We understand the reviewer's comment. All-in-all, we believe that while early POSS at 3-month might be common, persisting POSS at 6-month is not common and is problematic, especially if leading to deviations in routine treatment. Prof. Audigé also reported in 2015 (1), that while early POSS events might not be seen as a complication by all authors and surgeons, patient's perspective might be of interest. For all reported events in the ARCR_Pred study, we asked treating surgeons and patients to describe the perception of the severity of the experienced. We provide preliminary data below for POSS, the severity perception differed significantly ($p < 0.001$) between treating surgeons (mean (SD): 64 (25)) and patients (29 (10)).

Figure: Difference in severity perception between patients (red) and treating surgeons (blue) for post-operative shoulder stiffness

In the paper the reviewer mentioned (2), authors reported that the occurrence of early post-operative shoulder stiffness (POSS) (3-month) is associated with lower preoperative functional scores. This finding is supported by the findings of the present manuscript. Whether POSS is a factor protecting healing is not the scope of the present analysis, however early OSS is one factor that we are considering into the development of a prediction model for repair integrity.

Suggested changes: Lines 90-93: **“The occurrence of an early POSS is often perceived as a negative event by patients, which limits them in performing daily life activities, leading to prolonged rehabilitation or, in severe cases, to capsular release.”**^{5,6}

Methods

Reviewer #2: Lines 107-111. Who did the scoring, a person not otherwise involved in treatment or clinical follow-up?

Authors response: Clinical examinations were performed by treating surgeons or study staff, depending on the study center: assisting surgeons or study nurses.

Suggested changes: Lines 112-114: “Patient clinical examinations at baseline, 6 and 12 months were performed **by the treating surgeon or study staff** and included shoulder pain, active and passive range of motion and strength (Constant Score).”

Reviewer #1: Line 122: recommend adding description about the details of surgeon (surgical number per year, experience, etc.), which may also be added into multivariable regression analysis.

Authors response: Depending on the study center, the number and experience of surgeons varied. We plan to describe extensively the distribution of surgeon-level characteristics and its association with outcomes in a separate manuscript. Of note (see eTable 1), the 53 surgeons did not highlight the need to include surgeon experience in the present manuscript as a potential factor affecting stiffness. Adding this factor would require going through a new Delphi process, which we would like to avoid at that stage. We can however open the discussion on that point, and we provide the reader with more details towards the distribution of surgeon-level characteristics in the Supplement.

Suggested changes: Lines 122-123: “**The distribution of surgeon-level characteristics is also described in more details in eMethods.**”

Reviewer #1: Line 162-165: Do you perform the sensitivity analysis and how much data were missing? You should perform sensitivity analysis to make sure the imputation is correct.

Authors response: We realized now there was a mistake in the manuscript. We actually performed the whole analysis on complete and multiple imputed data. We would like to apologize for the confusion. All details are in the eMethods section.

Suggested changes: Line 165: “**Additional data management and analyses steps are described in eMethods.**” Lines 169: “**We performed and reported the whole analysis using complete case and multiple imputed data.**”

Reviewer #2: Line 172: AUC and Calibration are good and recommended indicators. Other studies on predictive models also use the Brier score as well. This way you get a better impression of the overall performance of the predictive model (see fx Steyerberg EW, Vickers AJ, Cook NR, Gerds T, Gonen M, Obuchowski N, et al. Assessing the performance of prediction models: a framework for traditional and novel measures. *Epidemiology*. 2010 Jan;21(1):128–38.). Adding the Brier score could thus ease comparison with other models. This is just a suggestion and not mandatory to accept the manuscript for publication.

Authors response: Brier score is indeed a proper overall performance metric. In a comprehensive review from Prof. Ben van Calster et al., authors suggests this metric is closely related to the R2 (preprint) (3). We believe reporting R2 is enough to describe the overall performance of the models at that stage and we would like to avoid reporting too many performance metrics.

Suggested changes: None.

Reviewer #2: Line 197: A brief description of the Lasso method would enhance easy understanding.

Authors response: We agree with the reviewer.

Suggested change: Lines 202-208: “Two variable selection procedures leading to a seven and eight-parameter models were compared: “expert” model (the seven parameters with the highest predictive ability, as reported in the Delphi survey) and least angle selection and shrinkage operator (“Lasso”) model. In our case, we used Lasso as a variable selection procedure.¹⁷ **Lasso is a linear regression method that automatically selects important variables by shrinking the coefficients of less important ones towards zero. In short, the aim of the variable selection procedure was to identify the most important prognostic factors associated with the outcome.** We combined the information from both Expert-model and Lasso-model and identified the “combined” model with seven parameters after (1) dropping variables that contained redundant information, (2) identifying the set of variables maximizing bootstrap validated performance metrics.”

Reviewer #3: Transparency of Parameter Selection. Question: Although the article mentions using methods such as Delphi surveys and Lasso regression to select parameters, there is a lack of detailed explanation for the final inclusion of certain parameters. For example, in Table 3, which presents the core results, the statistical significance of the "age" and "operation duration" parameter are above 0.05, yet it is still included in the model. This may make it difficult for readers to understand the reason for its inclusion. Suggestion: The authors should provide a more detailed account of the parameter selection process and the rationale for the final inclusion in the model.

Authors response 1: We agree with the reviewer that the related section can be confusing and left room for improvement. Of note, we applied two different variable selection procedures leading to one Expert-model and one Lasso-model. These models led to two different sets of 7 factors, as highlighted in the Table below. We identified the “combined” model using these 14 variables. We (1) dropped the redundant information (e.g., preoperative shoulder stiffness), and (2) we maximized our performance metrics by minimizing the overfitting, which led us to the identification of the 7 variables. We did not apply any variable selection procedure according to p values. In Prof. Steyerberg’s book, it is mentioned Pages 209-210, that “non-significance does not mean there is evidence for a zero effect of a predictor”. It is also suggested that p-values do not bring much when the aim is to maximize performance metrics, which is what we did in our case. We are happy to provide below the reviewers with an overview of the variable selection procedure:

Variables	Lasso-model	Expert-model	Combined-model
Baseline shoulder stiffness		X	
Diabetes status		X	
Baseline Oxford Shoulder Score		X	X
Baseline level of depression and anxiety		X	
Performance of a capsulotomy		X	
Symptom duration		X	X
Tear etiology		X	
Acromiohumeral distance	X		X
Age at surgery	X		X
Number of anchors used	X		
Surgery duration	X		X
Baseline active abduction	X		X
Baseline passive abduction	X		
Baseline external rotation at 0-degree abduction	X		X

Suggested changes: “Lines 208-212: We combined the information from both Expert-model and Lasso-model, and identified the “combined” model with seven parameters **after (1) dropping variables that contained redundant information, (2) identifying the set of variables maximizing bootstrap validated performance metrics.**”

Reviewer #3: Additionally, the limitations of the parameter selection methods could be discussed, along with possible directions for improvement.

Authors response 2: With regards to the last part of the comment and adding the limitations of the variable selection procedure in the discussion, we will extend the related section.

Suggested changes: Lines 383-385: “More research identifying optimal prediction models for outcome prediction in surgical patients is needed. Of note, the variables suggested in the Delphi survey for the assessment of prognostic relevance were identified using the findings of our systematic review.⁴ While seeking the surgeons’ expertise for the initial variable selection was essential, the perspectives of patients and rehabilitation professionals were not considered at that stage. **Doing so may have resulted in a different variable selection for the prediction model. In addition, the final model relied on shoulder surgeon expert knowledge and Lasso variable selection procedure only. The Lasso, while effective for variable selection, may exclude variables with small but potentially clinically relevant associations¹⁷, hence leaving room for further methodological comparison across modern variable selection procedures, such as Bayesian predictive projection³⁷.**”

Reviewer #1: Line 216 were the thresholds only defined by experienced surgeons? Is there any study discussing about the classification?

Authors response: Yes, they were only defined by the experienced surgeons coauthors of the study. They partly relied on the baseline prevalence of the outcome to define their risk probability thresholds which we estimated to be 10%. 25% represent a 2.5 fold increase in the risk and therefore represent a high risk group.

Suggested change: Lines 226-228: “Four experienced surgeons (AL, AMM, CB, and PM) defined three a priori risk probability thresholds: 25%, 15%, and 10% **based on the knowledge that the overall risk of POSS is around 10%.**”

Reviewer #1: Line 242-245, why 63 patients account for 40%?

Authors response: The reviewer is right. We need to revise this figure. Of the 973 included patients, 111 patients had the outcome (13%). Among them, 67 patients (60%) met the first condition, and 63 patients (56%, ... not 40%) met the second condition, with 29 patients having both conditions.

Suggested changes: Lines 253-257: “Using the definition suggested by surgeons in the Delphi survey, 111 patients (13%) suffered from POSS, with 67 patients (60%) meeting the first condition (occurrence of symptomatic stiff shoulder leading to a modification in the routine treatment between three- and six-month), and 63 patients (**56%**) meeting the second outcome condition (persisting restrictions in range of motion at six-month).”

Results

Reviewer #2: Line 265: I think eTable 5 is highly interesting in the way the reader can follow the model development and the improvements in performance. Why not put it in the article, not only in Supplementary material?

Authors response: We welcome this suggestion and therefore put the eTable 5 in the manuscript changed the order of the respective tables.

Suggested changes: Tables order in the manuscript and in the Supplement.

Discussion

Reviewer #1: Table3- The final model included an OR of 1.05 ($p=0.12$) for the operation duration, but its clinical significance was not explained (e.g., whether the duration of the procedure reflected surgical complexity or operator experience)

Authors response: We agree with the reviewer comment and provided more details on that aspect in the discussion.

Suggested changes: “**Interestingly, the number of anchors used, and surgery duration emerged as a factor associated with worse outcome using our Lasso selection procedure. This finding is also supported by the findings of another study for repair integrity outcome.²² Surgery duration is likely a proxy variable reflecting its association with other factors that influence the occurrence of POSS.**”

Reviewer #2: Lines 104-105: patient inclusion and operations were during the Covid pandemic. It is possible that this may have influenced patient selection for operative treatment, thereby possibly influencing the external validity of the results. Could patients more prone for stiffness be less represented in this cohort? This could be brought up in the discussion.

Authors response: We agree with the reviewer’s comment. It might be that the treated patients over that period were more severely affected by their condition than otherwise. Interestingly, we have had a look at the selection bias of the patients included in the ARCR_Pred study in our cohort description we recently published in PLOS One (4). However, we did not compare the distribution of the baseline characteristics and outcome profiles of our data to historical databases to assess the impact of the COVID pandemic to access to orthopedic procedures during that period. We suggest adding one sentence to the discussion to approach that topic.

Suggested changes: Lines 378-384: “**While we assessed the selection bias of the ARCR_Pred study elsewhere³¹, study patients were recruited and operated during the COVID-19 pandemic. This specific patient selection might limit the external validity of the ARCR_Pred-POSS model, which would remain to be assessed in an external cohort.**”

Reviewer #2: Good discussion, especially bringing up Bayesian methods and machine-learning methods. I am curious about what the authors think are the main reasons for the improvement in model performance, from the original KWS-POSS model to this final model. What is the contribution of the larger sample size and the expert consensus?

Authors response: We would like to thank the reviewer for their nice comment. Of note, the dataset used in the initial version of the KWS-POSS model was bigger (N = 1330 ARCR cases) (5). We have

three reasons that came to our mind: 1/ **patient population**: the KWS-POSS model was developed using a single private clinic in Zürich. The ARCR_Pred-POSS model was developed using a nationwide cohort study involving 19 study centers hence maybe limiting the ability of the KWS-POSS model to predict accurately in the whole cohort; 2/ **set of available prognostic factors**: new prognostic factors are only available in the ARCR_Pred study database, such as acromiohumeral distance, which might explain a part of the observed differences; 3/ **outcome definition** also slightly varied between the two studies, since the definition was revised by surgeons in the meantime and agreed on the fact that persisting restrictions in passive range of motion at 6-month should be characterized as a POSS, which is what we did in this new study.

Reviewer #3: Explanation of the Age Parameter. **Question:** The model includes the "age" parameter, which has an OR value of 0.98 ($p = 0.059$), indicating that the risk of POSS slightly decreases with each additional year of age. This result contradicts clinical intuition, as it is generally believed that older age is associated with a higher risk of POSS. The authors need to provide a detailed explanation for this finding. **Suggestion:** Discuss the potential physiological and pathological mechanisms underlying the relationship between age and POSS risk. For instance, could it be that older patients have weaker inflammatory responses or better rehabilitation compliance, thereby reducing the risk of POSS?

Authors response: Our finding is supported by the results of two studies included in our systematic review of prognostic factors for post-operative shoulder stiffness (6). In 2022, we discussed: "Of note, older age (more than 50 years old) was already found to be a protective factor for the occurrence of POSS [31]. Nevertheless, this association is still puzzling. On the one hand, older patients tend to have larger tears, for which repairs are thought to be prone to increased initial joint tightness [37], possibly also due to reduced initial tendon length [38]. Repairs of smaller, partial, and traumatic rotator cuff tears—which occur more frequently in younger patients, have been shown to be associated with a higher rate of POSS in previous studies [39]. The high incidence of preoperative rotator interval fibrosis observed in partial tears may partly explain this association [40]."

Suggested changes: Lines 339-344: "This finding is in line with the results of two studies included in our recent review, synthesizing the evidence related to prognostic factors for POSS.4 Repairs of smaller, partial, and traumatic tears—which occur more frequently in younger patients—have been shown to be associated with higher rates of POSS in previous studies.22 High incidence of preoperative rotator interval fibrosis observed in partial tears might partly explain this puzzling association.23 Additionally, younger patients are more often affected by traumatic lesions, which are associated with early intramuscular edema formation. Recent experimental studies have demonstrated that this trauma-related edema leads to muscle fiber disruption and progressive interstitial fibrosis, which may also contribute to the higher rates of postoperative shoulder stiffness observed in this population.24"

Reviewer #3: Rationality of Follow-up Duration. **Question:** The study followed up for 2 years, but the main objective was to predict the risk of POSS within 6 months after surgery. Although a 2-year follow-up can provide more comprehensive data, it may increase the complexity and cost of the study, and its contribution to the primary goal (predicting POSS risk within 6 months) is not clear. **Suggestion:** The authors should provide a detailed discussion on the necessity of a 2-year follow-up, including plans for utilizing the long-term data and its potential clinical significance. If the 2-year follow-up is mainly to capture long-term complications, this should be clearly stated in the discussion section, along with suggestions on how to use this data to further optimize clinical decision-making.

Authors response: The ARCR_Pred study was initiated to 1/ document the safety events up to two years after ARCR and 2/ develop prediction models for two primary outcomes: (a) post-operative shoulder stiffness within 6-month and (b) functional outcomes up to two years follow-up, using the Oxford Shoulder Score. The present manuscript aims to answer objective 1/ a) of the ARCR_Pred study. This does not preclude from performing an analysis on what is the adverse event status at last follow-up (24-month). It is recommended to complete follow-up over 2 years after ARCR (7) at least for a comprehensive documentation of all relevant adverse events.

Suggested changes: None.

Reviewer #3: Clinical Applicability of the Model. Question: Although the model performs well in terms of discrimination and calibration, the article lacks a detailed discussion on the clinical applicability of the model. For example, how can the model be applied in clinical practice, and how can clinical decisions be adjusted based on the model's results? **Suggestion:** Risk Probability Group Thresholds: In the discussion section, the authors should provide a detailed explanation of the rationale behind the thresholds for the three risk probability groups (0 - 10%: low risk, 10 - 25%: moderate risk, 25% - 100%: high risk), including whether they are based on clinical experience or data-driven analysis. Enhanced Clinical Applicability Discussion: Include a more detailed discussion on the clinical applicability of the model, such as its application in real clinical scenarios, potential clinical impact, and how it can be integrated with existing clinical guidelines. For example, for high-risk patients, are more aggressive preoperative rehabilitation plans or postoperative interventions recommended?

Authors answer: We agree with the reviewers' comment and we would like to refer to the answer we provided to Reviewer #2 in P. 6. The coauthors relied on the assumption that the overall population prevalence of POSS would be 10%. Clinicians involved in the present manuscript considered that patients with a 2.5-fold risk increase (25%) would be a threshold to consider for clinical applicability and changes in the routine treatment. We will add a few lines discussing this choice.

Authors answer: Lines 357-366: **“Our findings support a transparent and evidence-based treatment decision-making process, where individualized expected outcomes can be shared with patients.** We defined three risk probability groups (0 to 10%: low risk, 10 to 25%: moderate risk, and 25 to 100%: high risk). **For example, moderate and high-risk patients could be suggested alternative and preventive rehabilitation protocols, such as balneotherapy, recently shown to facilitate the early post-operative range of motion recovery²⁵. These risk probability thresholds were not unanimously agreed upon by all clinicians involved in the project. They should therefore be considered illustrative at this stage. Their relevance and applicability will be further assessed and validated once the prediction models are implemented in clinical practice.”**

References

1. Audigé L, Blum R, Müller AM, Flury M, Durchholz H. Complications Following Arthroscopic Rotator Cuff Tear Repair: A Systematic Review of Terms and Definitions With Focus on Shoulder Stiffness. *Orthop J Sports Med.* 2015;3(6):2325967115587861.
2. Takahashi R, Kawakami K, Harada Y, Kouzaki K, Kajita Y. Early Postoperative Stiffness After Arthroscopic Rotator Cuff Repair Correlates With Improved Tendon Healing. *Arthroscopy: The Journal of Arthroscopic & Related Surgery.* 2024;40(8):2186-94.
3. van Calster B, Collins GS, Vickers AJ, Wynants L, Kerr KF, Barreñada L, et al. Performance evaluation of predictive AI models to support medical decisions: Overview and guidance. 2024.
4. Stojanov T, Audigé L, Aghlmandi S, Rosso C, Moroder P, Suter T, et al. Baseline characteristics and 2-year functional outcome data of patients undergoing an arthroscopic rotator cuff repair in Switzerland, results of the ARCR_Pred study. *PLOS ONE.* 2025;20(1):e0316712.
5. Audigé L, Aghlmandi S, Grobet C, Stojanov T, Müller AM, Felsch Q, et al. Prediction of shoulder stiffness after arthroscopic rotator cuff repair. *Am J Sports Med.* 2021;49(11):3030-9.
6. Stojanov T, Modler L, Müller AM, Aghlmandi S, Appenzeller-Herzog C, Loucas R, et al. Prognostic factors for the occurrence of post-operative shoulder stiffness after arthroscopic rotator cuff repair: a systematic review. *BMC Musculoskeletal Disorders.* 2022;23(1):1-10.
7. Audigé L, Flury M, Müller AM, Durchholz H. Complications associated with arthroscopic rotator cuff tear repair: definition of a core event set by Delphi consensus process. *J Shoulder Elbow Surg.* 2016;25(12):1907-17.